# PET/CT and MR Improve Interobserver Agreement in Primary Tumor Determination for Radiotherapy in Esophageal Squamous Cell Cancer

**DOI:** 10.3390/diagnostics15060690

**Published:** 2025-03-11

**Authors:** Ajra Secerov-Ermenc, Primoz Peterlin, Vaneja Velenik, Ana Jeromen-Peressutti, Jasna But-Hadzic, Franc Anderluh, Barbara Segedin

**Affiliations:** 1Department of Radiation Oncology, Institute of Oncology Ljubljana, 1000 Ljubljana, Sloveniabsegedin@onko-i.si (B.S.); 2Faculty of Medicine, University of Ljubljana, 1000 Ljubljana, Slovenia

**Keywords:** squamous cell carcinoma, primary tumor, interobserver variability, positron emission tomography with computed tomography, magnetic resonance

## Abstract

**Background/Objectives**: The aim of the study was to evaluate interobserver variability in the determination of the primary tumor for radiotherapy treatment planning in esophageal squamous cell carcinoma (ESCC). **Methods**: Sixteen patients with locally advanced ESCC were included in the analysis. In all patients positron emission tomography with computed tomography (PETC/CT) and magnetic resonance (MR) scans for radiotherapy planning were performed. Five experienced radiation oncologists delineated the primary tumor based on CT alone, MR alone, PET/CT, CT with fused MR and PET/CT with fused MR. Mean tumor volumes were calculated for each patient and imaging modality. The generalized conformity index (CIgen) was calculated to assess agreement in tumor determination. **Results**: The mean tumor volumes and CIgen for CT alone, MR alone, PET/CT, CT with fused MR and PET/CT with fused MR were 33.1 cm^3^, 30.2 cm^3^, 38.1 cm^3^, 31.9 cm^3^, 36.2 cm^3^ and 0.59, 0.64, 0.66, 0.63, 0.71, respectively. CIgen was significantly higher using PET/CT with fused MR compared to CT (*p* < 0.001) and PET/CT (*p* = 0.002) and using PET/CT compared to CT (alone) (*p* = 0.003). **Conclusions**: Our study showed higher agreement in primary tumor determination in ESCC using PET/CT compared to CT alone. Higher agreement was also found using PET/CT with fused MR compared to CT alone and PET/CT.

## 1. Introduction

Imaging is essential for the diagnosis, staging, treatment planning, and post-treatment follow-up of esophageal squamous cell carcinoma (ESCC). Endoscopic ultrasound (EUS) is generally effective in evaluation of the clinical T stage of ESCC, though caution is needed when differentiating between Tis, T1a, T1b, and in T4 stages [1]. Computed tomography (CT) is commonly utilized for staging of ESCC; however, its accuracy in determining T stage remains limited [2].

Positron emission tomography (PET) combined with computed tomography (CT) is now established as a crucial diagnostic tool for the initial staging of ESCC. This is due to its high accuracy in detecting metastases, including pathological lymph nodes, with a sensitivity of 66% and a specificity of 96% [3]. The tracer most commonly used in ESCC diagnostics is 18-F-fluorodeoxyglucose (FDG), a glucose analog that accumulates in tissues with increased metabolic activity [4,5]. Another important application of FDG-PET/CT in the diagnosis of ESCC is the determination of the extent of the primary tumor. Studies have shown that there is a strong correlation between the tumor length measured on preoperative FDG-PET/CT scans and the actual dimensions in histopathological samples obtained after surgery [6,7]. However, several studies investigating interobserver variability in determining the primary tumor in radiotherapy have shown contradictory results. While some studies suggest potential benefits, others have shown no significant impact of FDG-PET/CT on improving interobserver agreement [8,9].

Magnetic resonance (MR) imaging is promising in the diagnostics of ESCC as it offers better resolution and soft tissue contrast compared to CT imaging. However, MR imaging of areas such as the upper abdomen or mediastinum is challenging due to motion artefacts caused by organ movement and the central location of these regions. In recent years, advances in technology have significantly minimized these imaging artefacts and improved overall image quality [10,11,12]. Hou et al. compared tumor lengths of ESCC determined by various imaging modalities, including CT, T2-weighted MR (T2-MR), and diffusion-weighted imaging (DWI), with pathological lesion length. Their results showed that DWI had a stronger correlation with pathological lesion length than CT or T2 MRI, making it the most accurate imaging method among the methods studied [13]. Moreover, Vollenbrock et al. concluded that agreement in determination of the primary tumor with MR is comparable to that with FDG-PET/CT [14].

Radiotherapy is a cornerstone in the preoperative and definitive treatment of non-metastatic esophageal cancer [15,16,17,18]. However, a recently published landmark study has shown that perioperative chemotherapy improves the survival of patients with resectable esophageal adenocarcinoma (EAC) compared to preoperative chemoradiotherapy [19]. Consequently, radiotherapy remains crucial for the treatment of ESCC only. Accuracy is fundamental in radiotherapy as it may impact local control and the incidence of toxic adverse events [20,21]. Uncertainties in delineation of the target and consequent deviations from the treatment protocol may lead to impaired local control [22]. The aim of our study was to analyze the role of PET/CT and MR on the determination of the primary tumor in radiotherapy for ESCC.

## 2. Materials and Methods

The study received approval from the National Medical Ethics Committee (Approval No. 0120-620/2019/3) on 21 January 2020, and was conducted in compliance with the Declaration of Helsinki. It was also registered in the ClinicalTrials.gov database under the identifier NCT05611658. Written informed consent was obtained from all participants prior to their inclusion in the study.

### 2.1. Patients

Previously, we had conducted a prospective analysis of 23 patients with carcinoma of the esophagus (adenocarcinoma and squamous cell carcinoma) [23]. For this analysis, we included data from patients with ESCC only. Patients had to fulfill the following inclusion criteria: locally advanced ESCC, planned preoperative or definitive chemoradiotherapy and no contraindications for MR imaging.

### 2.2. PET/CT

In all patients, an FDG-PET/CT scan was performed in the treatment position for radiotherapy using a Siemens Biograph™ mCT 40 PET-CT simulator (Siemens Healthineers, Erlangen, Germany) according to a standard preparation protocol. The administered activity of 18F-FDG was 3.7 MBq/kg via intravenous injection. Approximately 60 min later, a CT scan was performed with the following parameters: 120 kV, 200 mAs, a rotation time of 1 s, a pitch of 0.8 and a slice thickness of 3 mm. Before the CT scan, an iodine-based intravenous contrast agent was administered. Subsequently, a PET scan was performed in 3-dimensional mode with a duration of 2 min per bed position.

### 2.3. MR

MR was performed with a 1.5T Optima™ MR450w GE MR simulator (General Electric, Chicago, IL, USA). Patients were scanned in the treatment position before the start of radiotherapy, without the use of an intravenous contrast agent. T2 images in the transverse plane with a slice thickness of 3 mm and DWI were obtained for each patient.

### 2.4. Tumor Determination

Five radiation oncologists, each with more than five years of experience in the treatment of esophageal carcinoma, determined the gross tumor volume (GTV) using the Eclipse™ planning system (Palo Alto, CA, USA). To ensure a uniform approach, an initial meeting was held to familiarize the observers with MR images of esophageal carcinoma. Under the guidance of an experienced radiologist, they practiced the delineation of GTV on MR images on two pilot cases. The observers were given relevant information about the location and characteristics of the tumor. The GTV was contoured separately based on several imaging modalities: CT, PET/CT, MR, CT fused with MR, and PET/CT fused with MR. All study images were anonymized to maintain objectivity. GTV was defined as the tumor visible on imaging that encompassed the entire circumference of the esophagus without including regional pathological lymph nodes. To reduce the likelihood of recall bias, contouring was performed with different imaging modalities at least two weeks apart.

During contouring on PET/CT images, observers initially delineated the GTV on CT images and then corrected the volume as needed using PET images. The GTV_PET_ was defined as 20% of the maximum standardized uptake value (SUV). The PET and CT images were fused, and the visible tumor was contoured as GTV, with adjustments made as needed to include GTV_PET_ (Figure 1).

### 2.5. Data Analysis

GTV volumes were measured for each observer, each patient and all imaging modalities. Average volumes were then calculated for each patient and imaging modality. The generalized conformity index (CIgen) was calculated to assess agreement in tumor determination. This index is independent of the number of volumes analyzed. A CIgen of 1 means perfect agreement between the observers, while CIgen = 0 means that there is no overlap between the delineations [24].(1)$$CI\mathrm{gen}=\frac{\sum \mathrm{pairs}\;i\;j\left[Ai\;\cap Aj\right]}{\sum \mathrm{pairs}\;i\;j\left[Ai\;\cup Aj\right]\;},$$

The CIgen was determined for each patient and then averaged across all patients for each imaging modality.

For the original prospective trial, we determined the sample size based on data from existing studies. We calculated that approximately 21 patients would be required, with a significance level of 0.05 and a statistical power of 0.8 [9,14,25]. For the purpose of this analysis, we included only 16 patients with ESCC form the original group. The Wilcoxon signed rank test was used to analyze statistically significant differences. For statistical analysis, we used Statistical Package for the Social Sciences, version 29.0 (SPSS Inc., Chicago, IL, USA).

## 3. Results

A total of 16 patients diagnosed with ESCC and treated at our institute between April 2020 and May 2021 were included in this additional analysis. Thirteen were men and three were women; the average age was sixty-one years. The characteristics of the patients are presented in Table 1.

Tumor delineation using PET/CT imaging resulted in significantly larger mean volumes (38.10 cm^3^) compared to CT (33.06 cm^3^, *p* = 0.004). Delineations based on MR were significantly smaller compared to CT (*p* = 0.007). Volumes based on PET/CT with MR were significantly larger than on CT alone (*p* = 0.044) (Table 2).

CIgen was significantly higher using PET/CT MR compared to CT (*p* < 0.001) and PET/CT (*p* = 0.002). Statistically significant higher agreement was seen for PET/CT compared to planning CT alone (*p* = 0.003). No statistically significant difference was seen between CT alone and MR (*p* = 0.796) or CT fused with MR (*p* = 0.49) (Table 3).

## 4. Discussion

Our analysis showed that PET/CT significantly improves interobserver agreement in determining the primary tumor of ESCC compared to CT, and the agreement is even higher when MR is added. These results are in contrast to our previous analysis, which also included patients with EAC. The highest interobserver agreement was observed for PET/CT fused with MR, whereas CT alone showed the lowest agreement. However, only the difference in CIgen between MR alone and CT alone was marginally significant (*p* = 0.048) [23]. Interestingly, studies examining interobserver agreement in assessing the primary tumor, mainly in EAC, also found no significant differences between PET/CT, MRI, and CT. Schreurs et al. analyzed interobserver variation of target volumes between PET/CT and CT in majority of EAC and found no significant difference [26]. Nowee et al. also found no difference in the delineation of GTV between PET/CT and CT in six patients with esophageal cancer, three of whom had EAC and three of whom had ESCC [9]. Another study analyzing the contouring of GTV on MR and PET/CT images in six patients with esophageal cancer (three EAC, three ESCC) showed no significant differences [14]. On the other hand, Toya et al. analyzed 10 patients with only ESCC in their study. They compared the delineation of GTV using PET/CT and CT and found higher agreement with PET/CT (*p* = 0.005) [8]. We can assume that ESCCs are more avid tumors for FDG than EACs, so tumor assessment on PET/CT is more accurate. In addition, the majority of EACs are tumors of the distal part of the esophagus, which is more prone to movement, and the borders of the tumor are more difficult to assess even with various imaging techniques. Interestingly, MR improves the agreement of tumor assessment in ESCC even more when added to PET/CT. We can explain this by the fact that PET/CT improves the assessment of the location of the tumor, while MR improves the accuracy of tumor margin determination due to its superior soft tissue contrast.

To our knowledge, our study was the first to compare all three imaging modalities to determine interobserver variability. The fusion of all three modalities showed the highest agreement. This could be due to the fact that the observers obtained the most information about the characteristics and extent of the tumor from all three imaging modalities together and therefore determined the tumor with greater agreement. In contrast, in two other similar studies, only one imaging modality was compared with the other, without fusion, and consequently no improvement in agreement was observed [9,14].

Our study has some limitations. Firstly, we did not compare the extent of primary tumors on different imaging techniques with histopathological specimens, making it impossible to determine which imaging method was closest to the “ground truth”. Interestingly, there was only one study that examined primary tumors and regional lymph nodes with PET/MR, PET/CT, MR, and contrast-enhanced CT (CECT), with postoperative pathology serving as the reference standard for assessing diagnostic accuracy. PET/MRI was superior to PET/CT, MRI and CECT [27].

Secondly, our study involved fewer observers compared to two similar studies. The study by Nowee et al. involved 20 physicians from 14 different centers, whereas the study by Vollenbrock et al. involved 10 physicians from two centers [9,14]. As a result, our group was less heterogeneous and there was a possibility that some observers learned from others. This may limit the validity of our results and may not fully reflect the potential for adoption in routine clinical practice.

Thirdly, we analyzed data from only 16 patients. In the original prospective study, the sample was calculated for the analysis of patients with EAC and ESCC. In this “post hoc” analysis, we included data from patients with ESCC only, so the results may not be robust enough. Therefore, to confirm our results, a study with larger number of patients is warranted. However, two studies that used similar methods and imaging modalities both analyzed data from only six patients [9,14]. Neither study found a statistically significant difference in interobserver variability, which could be due to the fact that not enough real-world clinical scenarios were included.

Fourthly, the delineation of primary tumors began with CT alone, followed by MR alone, PET/CT, CT fused with MR, and finally PET/CT fused with MR. The poorest agreement was observed with CT, while the highest agreement was seen with PET/CT fused with MR. This could be partly due to the fact that the observers were able to memorize the anatomical details of the cases during successive contouring sessions, despite a minimum interval of 14 days—often longer—between sessions. Notably, the entire contouring process was completed within a period of almost two years.

Despite these limitations, we can conclude that we found significantly higher agreement in the determination of the primary tumor with PET/CT compared to CT alone, and that the agreement was even higher when we added MR to PET/CT. Interobserver variability could have an important clinical impact as it may influence the occurrence of toxic side effects and local control. Studies investigating the impact of contouring uncertainties on radiation dose distribution help us to identify which differences might be clinically relevant. Several studies on different tumors have shown that interobserver variability has a significant impact on dose coverage to the target or organs at risk [20]. For example, in the treatment of cervical cancer, the effect of interobserver variability on the dose distribution of MR-guided brachytherapy was studied. They found that uncertainty in delineation affected the dose distribution to the target by ±5 Gy and to the organs at risk by ±2–3 Gy [28]. In breast cancer, interobserver variability had no effect on target coverage, while the percentage of lung volume irradiated with 20 Gy (V20) varied between 5% and 25%, heart V20 varied between 0% and 7% and heart V10 varied between 2% and 20% [29]. We found no studies in the literature examining the effects of contouring uncertainties on the dosimetric parameters of radiation treatment plans for esophageal cancer. This could be a starting point for further research, as late toxic effects after radiation treatment for esophageal cancer may lead to poorer survival. The incidence of grade 3 or higher cardiovascular complications following radiotherapy for esophageal cancer is estimated to be between 10.8% and 16.3% and may be related to the dose received by cardiac structures, particularly the coronary arteries [30,31,32]. Pulmonary complications are also frequently observed after radiotherapy of thoracic tumors. In patients treated for esophageal cancer with neoadjuvant chemoradiation therapy and surgery, the mean lung dose was most strongly associated with postoperative pulmonary complications, which occurred in 22.1% of cases [33]. Both cardiovascular and pulmonary complications can have a significant impact on patient survival and quality of life.

Esophageal cancer is not a very common tumor, but it has high mortality and morbidity [34]. Appropriate treatment, including radiotherapy, is important to achieve local control of the disease, which may also have an impact on overall survival. Since higher interobserver agreement could potentially improve local control, it seems reasonable to additionally use MR and PET/CT in routine practice to facilitate the determination of the primary tumor for radiotherapy treatment planning. As esophageal cancer is not a common cancer, this would not present a major financial burden in most centers. However, it should be emphasized that optimization of imaging protocols and further research are needed.

## 5. Conclusions

In conclusion, our study demonstrated that PET/CT fused with MR achieved the highest agreement in identifying the primary tumor in ESCC. Considering the fact that ESCC is a relatively rare cancer, adopting PET/CT and MR for radiotherapy treatment planning and primary tumor determination seems feasible.

## Figures and Tables

**Figure 1 diagnostics-15-00690-f001:**
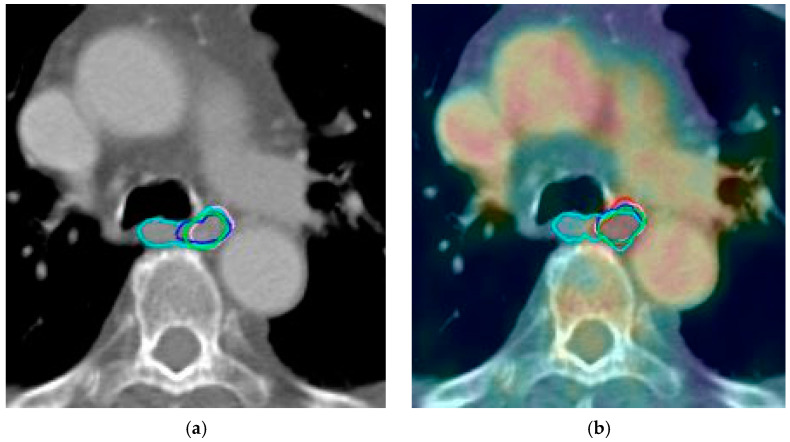
Green, cyan, red, purple, blue and magenta lines represent delineations of the tumor by each observer in case 9 based on: (**a**) computed tomography alone; (**b**) positron emission tomography with computed tomography; (**c**) computed tomography with fused magnetic resonance; (**d**) positron emission tomography with computed tomography and magnetic resonance.

**Table 1 diagnostics-15-00690-t001:** Characteristics of the patients with squamous cell esophageal cancer included in the study.

Case	Location—Third	Treatment	Stage
1	Proximal	definitive	T3N0M0
2	Proximal	definitive	T3N0M0
3	Middle	definitive	T3N1M0
4	Proximal	definitive	T3N2M0
5	Proximal	definitive	T3N1M0
6	Proximal	definitive	T3N1M0
7	Proximal	definitive	T3N0M0
8	Distal	preoperative	T3N1M0
9	Proximal	definitive	T3N0M0
10	Middle	preoperative	T3N0M0
11	Middle	definitive	T3N1M0
12	Proximal	definitive	T3N0M0
13	Proximal	definitive	T3N0M0
14	Proximal	definitive	T3N2M0
15	Middle	preoperative	T3N0M0
16	Middle	preoperative	T2N2M0

**Table 2 diagnostics-15-00690-t002:** Volume of tumor per imaging modality.

Case	CT (cm^3^)	MR (cm^3^)	PET/CT (cm^3^)	CT MR (cm^3^)	PET/CT MR (cm^3^)
1	20.3	16.0	17.2	18.5	17.7
2	33.2	33.2	33.4	35.2	32.5
3	113.4	105.0	114.4	110.9	115.7
4	20.2	18.0	30.3	17.7	28.2
5	29.1	18.1	35.2	22.1	31.8
6	22.8	13.7	26.6	23.8	24.8
7	117.3	109.7	123.2	112.5	114.7
8	36.9	24.7	55.0	30.4	37.7
9	36.1	33.7	49.7	25.5	50.5
10	22.6	17.2	22.7	20.4	21.9
11	2.2	2.6	3.4	3.0	3.5
12	14.9	16.2	24.0	17.6	21.6
13	20.0	16.7	25.6	19.0	24.9
14	19.4	16.8	19.4	18.5	19.0
15	8.3	28.5	17.6	22.2	23.5
16	12.2	12.8	11.6	13.1	11.5
AVG	33.1	30.2	38.1	31.9	36.2
SD	33.5	31.1	34.1	31.9	32.6

cm^3^ = cubic centimeter; CT = computed tomography; MR = magnetic resonance imaging; PET/CT = positron emission tomography and CT; CT MR = fusion of CT and MR; PET/CT MR = fusion of PET/CT and MR; AVG = average; SD = standard deviation.

**Table 3 diagnostics-15-00690-t003:** Generalized conformity index per imaging modality.

Case	CT	MR	PET/CT	CT MR	PET/CT MR
1	0.69	0.75	0.74	0.78	0.76
2	0.62	0.67	0.65	0.51	0.67
3	0.83	0.8	0.81	0.86	0.83
4	0.57	0.66	0.68	0.51	0.71
5	0.69	0.66	0.75	0.47	0.76
6	0.58	0.47	0.77	0.62	0.75
7	0.76	0.78	0.79	0.76	0.8
8	0.66	0.62	0.5	0.68	0.71
9	0.47	0.42	0.67	0.46	0.68
10	0.71	0.69	0.76	0.67	0.79
11	0.28	0.29	0.47	0.37	0.59
12	0.58	0.59	0.52	0.51	0.55
13	0.69	0.66	0.75	0.69	0.76
14	0.77	0.74	0.78	0.85	0.78
15	0.37	0.78	0.53	0.7	0.66
16	0.13	0.62	0.46	0.58	0.56
AVG	0.59	0.64	0.66	0.63	0.71
SD	0.19	0,14	0.13	0.15	0.09

CT = computed tomography; MR = magnetic resonance imaging; PET/CT = positron emission tomography and CT; CT MR = fusion of CT and MR; PET/CT MR = fusion of PET/CT and MR; AVG = average; SD = standard deviation.

## Data Availability

The original contributions presented in this study are included in the article. Further inquiries can be directed to the corresponding author.

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
