# Peer review of "PET/CT and MR Improve Interobserver Agreement in Primary Tumor Determination for Radiotherapy in Esophageal Squamous Cell Cancer"

_diagnostics, 2025, doi:10.3390/diagnostics15060690_

Round 1
Reviewer 1 Report
Comments and Suggestions for Authors
This study evaluates the impact of PET/CT and MR imaging on interobserver variability in delineating the primary tumor for radiotherapy in ESCC. The results suggest that PET/CT improves agreement compared to CT alone, and that PET/CT fused with MR yields the highest interobserver agreement. The study is well structured. However, several points need improvement.
Main comments:
1. The discussion provides a general overview of previous research, but lacks a critical analysis of why the results of this study differ from previous findings. Some previous studies found no significant effect of PET/CT on interobserver agreement, but the authors do not explain why their study had different results. Given the small sample size (16 patients) and the lack of a histopathologic gold standard, it remains unclear what unique contribution this study makes to the field. The discussion should explicitly compare the study's methodology with previous research and clarify its added value beyond simply mentioning the existing literature.
2. The study demonstrates statistically significant improvements in interobserver agreement with PET/CT and MR, but does not discuss how this translates into clinical benefits such as improved treatment precision or patient outcomes. Increased agreement does not necessarily translate into improved accuracy or clinical decision making.
3. While the study uses the Wilcoxon signed-rank test for statistical comparisons, there is limited discussion of whether the sample size provides sufficient power for these tests. A more robust discussion of statistical power and potential alternative methods would be beneficial.
Minor comments:
4. While the study concludes that PET/CT fused with MR provides the highest agreement, it does not discuss whether this approach is feasible in routine clinical practice. Addressing factors such as cost and accessibility would be beneficial.
Author Response
Review 1
This study evaluates the impact of PET/CT and MR imaging on interobserver variability in delineating the primary tumor for radiotherapy in ESCC. The results suggest that PET/CT improves agreement compared to CT alone, and that PET/CT fused with MR yields the highest interobserver agreement. The study is well structured. However, several points need improvement.
Main comments:
- The discussion provides a general overview of previous research, but lacks a critical analysis of why the results of this study differ from previous findings. Some previous studies found no significant effect of PET/CT on interobserver agreement, but the authors do not explain why their study had different results. Given the small sample size (16 patients) and the lack of a histopathologic gold standard, it remains unclear what unique contribution this study makes to the field. The discussion should explicitly compare the study's methodology with previous research and clarify its added value beyond simply mentioning the existing literature.
Thank you for pointing this out. We agree with the comment and have therefore emphasized the differences between our study and previous ones. To our knowledge, we are the first to compare tumor delineation on all three imaging modalities together. The fusion of all three modalities showed the highest agreement. We have added a paragraph discussing this point (lines 204 – 210). In discussion there already was a section on studies which included patients with adenocarcinomas and resulted in poorer agreement. Adenocarcinomas are located in the distal part of the esophagus, which is more difficult to assess. Moreover, adenocarcinomas are usually less avid for FDG than squamous cell carcinomas. This is now addressed with more emphasis (lines 196 – 203).
- The study demonstrates statistically significant improvements in interobserver agreement with PET/CT and MR, but does not discuss how this translates into clinical benefits such as improved treatment precision or patient outcomes. Increased agreement does not necessarily translate into improved accuracy or clinical decision making.
We appreciate your suggestion regarding the potential clinical benefit of improved interobserver agreement and we have added a paragraph and revised the introduction accordingly (lines 63 – 70; lines 241 – 264). Interobserver variability could have an impact on local control and incidence of toxic adverse event. Several studies have demonstrated that uncertainties in delineation of the target and consequently deviations from the treatment protocol may lead to impaired local control.
- While the study uses the Wilcoxon signed-rank test for statistical comparisons, there is limited discussion of whether the sample size provides sufficient power for these tests. A more robust discussion of statistical power and potential alternative methods would be beneficial.
Thank you for your valuable comment. We have calculated the sample size for original prospective study, which included both patients with squamous cell carcinoma and adenocarcinoma. For this “post hoc” analysis, we only included patients with squamous cell carcinoma, so the analysis may be underpowered. We have added a paragraph to the manuscript to clarify this issue (line 141 – 145). Interestingly, two studies that used a similar methodology analyzed an even smaller number of patients (only 6).
Minor comments:
- While the study concludes that PET/CT fused with MR provides the highest agreement, it does not discuss whether this approach is feasible in routine clinical practice. Addressing factors such as cost and accessibility would be beneficial.
We agree with your suggestion and have made the necessary changes. We have added a paragraph explaining the potential cost benefit of using additional imaging techniques (line 265 – 272).
Thank you for your constructive comments. They have helped us to improve the quality of the paper. We hope the revisions meet your expectations and we look forward to your feedback.
Reviewer 2 Report
Comments and Suggestions for Authors
This article focused on the significance of PET/CT and MR on improving interobserver agreement in primary tumor determination. The authors firstly introduced the fact that present studies investigating interobserver variability in determining the primary tumor have shown contradictory results and thus demonstrated the importance of the study. Most importantly, despite arriving at conclusions that contradict some previous studies, the author provides a detailed and well-reasoned explanation for the possible reasons behind these discrepancies.
However, my primary concern is this: What do the results presented in Table 1 regarding tumor delineation between PET/CT vs. CT or MR vs. CT actually indicate, and what practical significance do they hold? Furthermore, as the authors themselves noted, the study "did not compare the extent of primary tumors on different imaging techniques with histopathological specimens, making it impossible to determine which imaging method is the closest to the ground truth." Given this limitation, does focusing solely on inter-observer consistency carry less substantial meaning? It would be more compelling if it could be demonstrated that the best method of tumor delineation showed the greatest interobserver agreement.
Author Response
Review 2
This article focused on the significance of PET/CT and MR on improving interobserver agreement in primary tumor determination. The authors firstly introduced the fact that present studies investigating interobserver variability in determining the primary tumor have shown contradictory results and thus demonstrated the importance of the study. Most importantly, despite arriving at conclusions that contradict some previous studies, the author provides a detailed and well-reasoned explanation for the possible reasons behind these discrepancies.
However, my primary concern is this: What do the results presented in Table 1 regarding tumor delineation between PET/CT vs. CT or MR vs. CT actually indicate, and what practical significance do they hold?
Thank you for your valuable comment. Indeed, the data in Table 1 have no direct impact on the results of the study. However, they illustrate the characteristics of the patients and give more confidence in the analysis as they represent real-world data. If it is still unnecessary in your opinion, we can delete it.
Furthermore, as the authors themselves noted, the study "did not compare the extent of primary tumors on different imaging techniques with histopathological specimens, making it impossible to determine which imaging method is the closest to the ground truth." Given this limitation, does focusing solely on inter-observer consistency carry less substantial meaning? It would be more compelling if it could be demonstrated that the best method of tumor delineation showed the greatest interobserver agreement.
Thank you for pointing this out. We agree with the comment and therefore we have added a paragraph explaining the clinical benefit of good interobserver agreement (lines 63 – 70; lines 241 – 264). Interobserver variability could have an impact on local control and incidence of toxic adverse event. Several studies have demonstrated that uncertainties in delineation of the target and consequently deviations from the treatment protocol can lead to impaired local control.
On the other hand, studies comparing the extent of squamous cell esophageal carcinoma with histopathological specimen are rare. Two studies compared tumor size measured on PET/CT and one measured on MR with pathological specimen. (Han, Zhong, Hou) There was only one study that examined primary tumors and regional lymph nodes with PET/MR, PET/CT, MR, and contrast-enhanced CT (CECT), with postoperative pathology serving as the reference standard for evaluating diagnostic accuracy. (Wang)
Thank you for your valuable feedback and insights. We hope the revisions meet your expectations, and we look forward to your feedback.
Round 2
Reviewer 1 Report
Comments and Suggestions for Authors
I confirm all concerns have been addressed.
Reviewer 2 Report
Comments and Suggestions for Authors
I have reviewed the revised manuscript and the authors' reply to my previous review comments. The authors have made improvements to the manuscript, particularly in the clarification of the meaning/significance of their result. I believe this article will help readers to learn the significance of PET/CT and MR in improving interobserver agreement in esophagus primary tumor determination, and make more progress in this field.